# Exploring New Routes for Genetic Resistances to Potyviruses: The Case of the *Arabidopsis thaliana* Phosphoglycerates Kinases (PGK) Metabolic Enzymes

**DOI:** 10.3390/v14061245

**Published:** 2022-06-08

**Authors:** Mamoudou Diop, Jean-Luc Gallois

**Affiliations:** 1Unité de Génétique et Amélioration des Fruits et Légumes (GAFL), INRAE, 84140 Montfavet, France; mamoudou.diop@inrae.fr; 2Biologie Santé, Biologie Végétale Marseille Luminy, Aix Marseille Université, F-13009 Marseille, France

**Keywords:** virus, genetic resistance, metabolic enzyme, *Arabidopsis*, PGKs, susceptibility genes, genome editing

## Abstract

The development of recessive resistance by loss of susceptibility is a consistent strategy to combat and limit damages caused by plant viruses. Susceptibility genes can be turned into resistances, a feat that can either be selected among the plant’s natural diversity or engineered by biotechnology. Here, we summarize the current knowledge on the phosphoglycerate kinases (PGK), which have emerged as a new class of susceptibility factors to single-stranded positive RNA viruses, including potyviruses. PGKs are metabolic enzymes involved in glycolysis and the carbon reduction cycle, encoded by small multigene families in plants. To fulfil their role in the chloroplast and in the cytosol, PGKs genes encode differentially addressed proteins. Here, we assess the diversity and homology of chloroplastic and cytosolic PGKs sequences in several crops and review the current knowledge on their redundancies during plant development, taking *Arabidopsis* as a model. We also show how PGKs have been shown to be involved in susceptibility—and resistance—to viruses. Based on this knowledge, and drawing from the experience with the well-characterized translation initiation factors eIF4E, we discuss how PGKs genes, in light of their subcellular localization, function in metabolism, and susceptibility to viruses, could be turned into efficient genetic resistances using genome editing techniques.

## 1. Introduction

Viruses represent a growing threat for agronomical production and cause important economic losses. Many strategies must be elaborated to both combat and limit damages from these pathogens, and developing genetic resistances is one of them [1,2]. To achieve a successful infection, viruses recruit plant factors in order to replicate and move from cell to cell throughout the plant. The characterization of those factors, called susceptibility factors, is of special interest to the breeder: plants whose susceptibility factors are modified and can no longer be hijacked by a virus are resistant to this virus [3]. A number of those resistances, often characterized as recessive, have been identified so far, especially resistance to RNA viruses, including the large group of potyviruses. Natural recessive resistance genes based on modification or alteration of host factors used by viruses are well-documented in the literature. *rym11*, *cmv1*, *zym*, *rymv2*, *bc-2*, and *bc-4* are natural recessive alleles conferring resistances against RNA viruses in important crops [4,5,6,7,8,9,10,11]. Some of these have been characterized based on traits that have long been selected during crop domestication, before the underlying genes and mechanisms were unravelled. Knowledge on their function and the basis for resistance is useful, as this can help to translate these functions to other crop species [12]. Host susceptibility factors involved at several step of the virus cycle have now been identified by different approaches, such as eIF4E, eIF4G factors, TOM1, TOM2, and EXA1 in the model *Arabidopsis* [13,14,15,16,17]. The translation initiation factors eIF4Es represent the archetypal example of these recessive resistances [18,19]. Natural variant and modified eIF4Es-based genetic resistances exist and are conceptual proof that a plant pro-viral factor can be transformed into resistance. Originally being the basis of many resistances selected during crop domestication, eIF4E factors have gathered a lot of attention as targets for biotechnology development, giving rise to questions about the most efficient way to develop wide-spectrum and durable resistances without compromising the plant development and yield [13,18,20,21].

Of those susceptibility factors, genes encoding metabolic enzymes have long been known as important factors for RNA viruses [22,23]. Phosphoglycerate kinases (PGKs) belong to this category of host enzymes with pro-viral activity. PGKs are metabolic enzymes involved in carbohydrates synthesis and energy production in prokaryotes and eukaryotes. Moreover, beyond their role in metabolism, PGKs have been shown to be essential for resistance to single-stranded positive RNA (ssRNA+) viruses [22,24]. Here, we focus on PGKs for their particular interest as a source of recessive resistant genes, expecting that PGKs could fodder plant breeding approaches in many crop species toward which ssRNA+ viruses are a major threat. At the same time, investigating the role of PGKs in crops resistance is of interest because it poses challenges that have already been addressed for resistance by loss of susceptibility, especially for *eIF4Es* genes, including gene redundancy and the trade-off between resistance versus plant development and yield. Finally, the fact that PGKs are enzymes required in both the cytosol and the chloroplast extends the difficulty of assessing their roles across different subcellular localizations.

Here, taking *Arabidopsis thaliana* as a model, we review current data available on the PGKs enzymes, with a special focus on the chloroplastic PGK2, whose involvement in the potyviruses cycle is documented [24,25]. We highlight structural homologies between the *Arabidopsis* PGKs in both primary and tertiary structures and conduct a phylogenetic analysis of PGKs from different organisms confirming the close relationship between cytosolic and chloroplastic forms found in plants. We also confirm the existence in plants of a more divergent group of PGKs harbouring a predicted methyltransferase domain (MT domain). Based on current knowledge on PGKs as susceptibility factors to viruses, we discuss the possibilities of finding and validating regions or amino acid residues in PGKs that are essential to produce resistance. We also discuss how targeting these regions in crops by using a genome editing strategy could allow the development of new genetic resistances against viruses.

## 2. *Arabidopsis* Encodes Two Types of Homologous PGKs Differentially Localized inside the Cell

Glycolysis consists of an enzymatic breakdown of glucose to form pyruvic acid. In plants, it takes place in both the cytosol and chloroplast, so glycolytic enzymes have to be addressed in both compartments. Indeed, the first plant PGKs sequences characterized in wheat (*Triticum aestivum*) correspond to two genes encoded by the nuclear genome, but expressing proteins that are differentially targeted inside the cell: one of the two genes was shown to display an N-terminal sequence extension coding for a transit peptide (TP), a type of sequences known to target the nucleus-encoded proteins to the chloroplast after their cytosolic translation [26]. The *Arabidopsis* genome contains three PGKs genes: *PGK1* (At3g12780) and *PGK2* (At1g56190), which encode proteins harbouring an N-terminal TP for chloroplast targeting (Figure 1, in lower cases), and *PGK3* (At1g79550), which lacks the TP-coding sequence and encodes a cytosolic enzyme [27,28].

The subcellular localisation of these three enzymes was confirmed by stable expressions of PGK-GFP translational fusions in *Arabidopsis*: PGK1 and PGK2 are localized in plastids, while PGK3 is expressed in the cytosol and in the nucleus [27] (Figure 1). To avoid any confusion due to the PGKs nomenclature inconsistency in previous reports, we will name here the *Arabidopsis* isoforms targeted to chloroplast as cPGK (hence cPGK1 and cPGK2) and the cytosolic version as a plain PGK (PGK3).

The *Arabidopsis* chloroplastic and cytosolic PGKs are highly similar (Figure 1). cPGK1 shares 91% and 84% amino acid identity with cPGK2 and PGK3, respectively. Without the TP, the amino acid identity between both chloroplastic forms (cPGK1 and cPGK2) increases to 94%. Interestingly, the plant PGKs isoforms have been shown to be phylogenetically closer to the bacterial PGKs than to other eukaryotic PGKs [29]. The *Arabidopsis* PGKs isoforms share around 60% of protein sequence identity with the *Geobacillus stearothermophilus* PGK, a PGK whose 3D-structure has been solved (PDB ID: 1PHP). This homology is comforted by the strong conservation of residues involved in adenosine triphosphate (ATP) and 3-phosphoglycerate (3-PGA)-binding sites in their protein sequences (red and black boxes, respectively) (Figure 1).

## 3. Phylogenetic Studies Confirm the Existence of a Major Clade of PGK, Grouping Cytosolic and Chloroplastic Forms, While Identifying a New Clade of Atypical PGKs

To confirm the generality of the existence of cytosolic and chloroplastic PGK in plants, as well as to assess gene redundancy, PGKs from agronomically important crops were retrieved. Here, we chose the crops from five distinct families for which ssRNA+ viruses represent a major threat. PGKs from *Homo sapiens*, *Saccharomyces cereviseae*, *Chlamydomonas reinhardtii*, and *Geobacillus stearothermophilus* were added as distant homologs. Sequences of 53 PGKs of plant were aligned in order to identify conserved residues and the presence of TPs (Appendix A). Duplicate sequences were removed. This resulted in 43 sequences, including 13 cytosolic forms, 24 chloroplastic forms, and 6 PGKs with a different structure harbouring a homologous methyltransferase (MT) domain (Appendix A). Such atypical PGKs have previously been described [30]. A PGK of *Klebsormidium nitens*, a streptophyte algae, was retrieved in GenBank, as the most similar non-plant sequence to this plant MT-containing PGKs (with 90% of coverage), and added to the set of sequences (Appendix A). Finally, 55 sequences were retained for phylogenetic analysis, including the 43 plant sequences (Appendix A). 

To look at the evolutionary relationship between plant PGKs, we built a phylogenetic tree using the maximum likelihood approach. Plant PGKs are clustered in two groups, group I and group II, supported by strong bootstraps of 99 and 100, respectively (Figure 2A). All three *Arabidopsis* PGKs are located in group I, which can be further divided into two subclades (with branches represented in green and blue) corresponding to chloroplastic and cytosolic proteins, respectively. cPGK1 and cPGK2 grouped with the chloroplastic forms, while PGK3, as expected, clustered with the plant cytosolic PGKs (Figure 2A). The PGKs group I is closely associated with the *Chlamydomonas reinhardtii* cPGK, as previously pointed out by Brinkmann and Martin [29]. This supports the hypothesis that plant chloroplastic and cytosolic forms both originate from a cyanobacterial ancestor [29].

The PGKs group II contains six atypical PGKs proteins from Poaceae, Cucurbitaceae, and Solanaceae (Figure 2A). Members of this group, which includes the aforementioned retrieved *K. nitens* cPGK, carry a conserved MT domain (Figure 2B) and are poorly homologous to group I PGK: their amino acid sequences show less than 30% of identity with the group I, and key residues of the enzymes involved in ATP and 3-PGA binding are absent. They correspond to a class of nuclear PGKs that have recently been described in rice (*Oryza sativa*), maize (*Zea mays*), grapevine (*Vitis vignifera*), and sorghum (*Sorghum bicolor*) [30]. Given the homology of structure, it is likely that these PGKs group II harbouring an MT domain may originate or share a common ancestor with algae PGK, such as *K. nitens* (Figure 2C). Interestingly, streptophyte algae are assumed to be the ancestors of land plants [31,32]. Those genes were retained in plant families like Solanaceae, Cucurbitaceae, and Poaceae, but are absent from Brassicaceae, including *Arabidopsis* and the Asteraceae families. The absence in these families of orthologous group II PGKs could be caused by gene loss during evolution. As pointed out by Massange et al. [30], the function of the atypical PGK-MT is intriguing and would need further characterization. Their predicted nuclear localization could make them candidates for involvement as part of DNA-polymerase-α complexes [33]. For the rest of the study, we will focus on the group I PGKs, to review their functions in the chloroplast compartment, and in the cytosol of higher plants.

## 4. 3D Protein Models of *Arabidopsis* PGKs Confirm the High Similarity between Cytosolic and Chloroplastic Forms

The obtention of the PGKs 3D protein structures by X-ray crystallography were instrumental to understand the PGKs organisation and enzymatic domains. The first protein 3D structures were characterized for the yeast (PDB ID: 1FW8) and the horse PGKs [34,35]. Both structures are similar in terms of described domains: the enzyme N-terminal and C-terminal domains contain two separate substrate sites that bind 3-PGA and ATP, respectively [35,36]. Here, we selected the *G. stearothermophilus* PGK, a close homolog to *Arabidopsis* PGKs—around 60% of sequence homology—and for which a 3D structure is available (PDB ID: 1PHP) as a reference model for prediction. Using its structure as a template, protein 3D models were built for all three *Arabidopsis* PGKs using the Swiss-Model work space (https://swissmodel.expasy.org/interactive, accessed on 28 July 2021).

The obtained models are presented in Figure 3. In accordance with the high homology between the PGKs amino acid sequences, these models are very similar and show the presence of two domains, as described above: a 3-PGA binding site at the N-terminal and an ATP binding domain at the C-terminal (Figure 3E) can be pinpointed because of the amino acid differences locally affecting the small α-helices. cPGK1 presents an extra α-helix at V82-G83-D84 that is not present in cPGK2 (compare Figure 3A, white arrow, and 3B, as well as the overlap in 3D). Conversely, cPGK2 presents an extra helix formed by A360-S361-A362 residues (Figure 3B) that is not present in cPGK1 (Figure 3A, orange arrow). Two dissimilarities are also observed when comparing cPGK2 and PGK3 (Figure 3F, white and orange arrow in C, and white boxes in F). Although the described differences impact the protein backbones, it remains to be tested whether these dissimilarities could change the enzymatic function in the cytosol or the chloroplast, or affect other properties (see below). To this aim, recent functional studies have been performed in *Arabidopsis* and have largely extended our knowledge on the roles of PGKs in plant metabolism and on their redundancies.

## 5. PGKs Fill Central Roles in Plant Metabolism and Present Both Specific and Redundant Functions

PGKs catalyse, during glycolysis, the transfer of phosphate from 1,3-BiPGA to ADP to form ATP and 3-PGA. In plants, PGKs are also required in the carbon reduction cycle. As a result, the involvement of the PGKs is expected at three stages of the plant metabolism: glycolysis in the cytosol, glycolysis in the chloroplasts, and the Calvin cycle in the chloroplasts (Figure 4). The plant PGKs are encoded by a few genes of varying number, depending on the species. However, it is unclear if redundant PGKs copies have specific or interchangeable roles in metabolism, especially in light of their differential subcellular targeting in the chloroplast and cytosol [27]. Therefore, one key is to understand the redundancy between PGKs and their possible functional complementation. *Arabidopsis* provides a good model to examine these aspects; the presence of a single cytosolic PGK3 points to a unique role in cytosolic glycolysis, but the presence of both cPGK1 and cPGK2 question their specificity or redundancy in the chloroplast compartment, in the Calvin cycle, or in glycolysis, respectively. To understand the specificity of each individual PGK, two teams carried out PGKs loss-of-function analysis using T-DNA insertion, RNAi, and CRISPR-Cas9 knockout alleles of these genes [27,38]. A summary of the phenotypes displayed by these mutant lines is given in Table 1.

In the analyses, the three single mutants did not display any severe phenotypes, although the plants knocked out for PGK3 displayed a significant reduction in development [27]. Notably, the previously reported albino phenotype associated with *cPGK2* loss-of-function [38] turned out to be independent from the *cPGK2* gene inactivation: RNAi downregulation, as well as CRISPR-Cas9-generated *cpgk2*^KO^ lines, were not significantly affected in development [27,38]. This suggests a global good redundancy between the three *Arabidopsis* PGKs, as well as regulatory mechanisms. For example, 3-PGA was found to increase in a *pgk3*^KO^ mutant, despite the absence of the cytosolic PGK3. Rosa-Tellez et al. [27] suggested that in the absence of PGK3, cPGK2 enhanced expression could lead to an increase in the amount of chloroplast 3-PGA, which can be transported to the cytosol by the triose phosphate transporter (TPT). However, analysis of the combined mutations unravelled the limits of this redundancy: an analysis on the recently obtained CRISPR-Cas9 mutant revealed that when both cPGK1 and cPGK2 are absent, the *Arabidopsis* plant displayed a lethal albino phenotype that could be partially reverted by adding an external carbohydrate. These results show that the function of the chloroplastic isoforms, cPGK1 and cPGK2, cannot be complemented by the cytosolic PGK3. Additionally, the cytosolic isoform, PGK3, was not able to complement the lethal phenotype after its chloroplast targeting. This observation suggests a possible divergent function between chloroplastic and cytosolic forms, despite their higher structural homologies [38]. It would be interesting to see whether the minor differences in the protein 3D structures noted above (Figure 3) could play a role in this functional divergence. In contrast, the *cpgk1* × *pgk3* lines displayed a wild-type phenotype [27]. Since the latter mutants carry only a functional cPGK2 gene, further investigations are needed to decipher the molecular mechanism that maintains plant metabolism in these lines. One of the possibilities is that the plant may produce different cPGK2 isoforms targeted to the chloroplast, but also to the cytosolic compartment through post translational modification or alternative splicing. It will also be of interest to generate the yet unavailable mutant combination, *cpgk2* × *pgk3*, to see if its phenotype mirrors the one displayed by the *cpgk1* × *pgk3* combination. What is interesting to note is that in addition to the structural homology described above, a functional homology between the cPGKs was unravelled. While cPGK2 on its own can support the plant development, PGK3 alone cannot fulfil the PGK’s function. It will be interesting to understand the reason underlying these differences: expression, cellular localization, enzymatic properties, and flux of metabolites. These effects on the plant metabolism will have to be taken into consideration when considering how these genes can be modified in plants in order to design genetic resistances. Interestingly, like the redundant eIF4E factors family used by ssRNA+ viruses to multiply, many glycolytic enzymes, including PGKs, have been identified as pro-viral factors to plant viruses.

## 6. PGKs Act as Host Pro-Viral Enzymes

Some proteins involved in metabolism have long been known to fulfil other functions in the cell, a fact known as protein moonlighting. This moonlighting activity could result from the protein adopting different cell type locations, different subcellular localizations, or acquiring the capacity to bind to different substrates [40]. For example, in humans (*Homo sapiens*), some of the PGK1 pool is localized in the nucleus, where it acts as a primer recognition protein (PRP), a cofactor of DNA polymerase-α [33]. Because they are abundant and evolutionarily constrained, glycolytic enzymes are also prone to be hijacked by viruses. PGKs have been found to fill other non-glycolytic functions during the virus cycle, and PGK extracted from the bovine brain was initially shown to be involved in mRNA synthesis of *Sendai virus* (SeV), a paramyxovirus. It was suggested that the glycolytic enzyme role was to stimulate the genomic RNA synthesis at the elongation step [41].

Recent investigations have also identified PGKs as susceptibility factors to RNA viruses in plants. The involvement of a chloroplastic PGK in plant virus multiplication was first described through the analysis of protein interactions between *Bamboo mosaic virus* (BaMV, potexvirus) and *Nicotiana benthamiana* proteins. The role of PGK in the BaMV cycle was confirmed by knocking down the chloroplastic PGK expression in tobacco [23]. Initially, it was not clear why a virus could recruit a chloroplast-localized protein, but subsequent study suggested that the BaMV directly recruits the glycolytic enzyme to access and replicate in the chloroplast [42] (as illustrated in Figure 5 for potyviruses, pathway I). This first observation points out the importance of PGKs for virus multiplication in tobacco leaves and supports chloroplast targeting as a required condition for virus multiplication (Figure 5) [42,43]. The outstanding question is the following: why does a virus need to replicate inside the chloroplast, a central organellar which is crucial for plant defence against pathogens?

The roles of PGKs as susceptibility factors were extended to the largest genus of ssRNA+ viruses, potyviruses, following the screening for natural resistances to *Watermelon mosaic virus* (WMV) among a panel of 53 *Arabidopsis* accessions. The *Arabidopsis Cabo verde island* (Cvi) accession was found to be resistant to WMV, while most accessions, such as *Landsberg erecta*, are highly susceptible to this virus. The Cvi recessive resistance was shown to be caused by a single allele, *rwm1*, that was mapped closely to *cPGK2*. While the Cvi resistance phenotype showed an incomplete penetrance effect (few Cvi plants are susceptible to WMV), the ectopic expression of the *Landsberg erecta* susceptible allele of cPGK2 in the Cvi accession restored the full susceptible phenotype to WMV, supporting a direct link between the resistance phenotype and *rwm1*. Moreover, complementary studies pointed to a serine–glycine substitution (S78G) in the cPGK2 protein of *Arabidopsis* Cvi ecotype [24]. This point mutation may alter or reduce the ability of the enzyme to interact with the viral proteins, or alter the protein subcellular localization or stability. In a twin study, a partial resistance to *plum pox virus* (PPV) was identified in Cvi by a genome-wide association analysis (GWAS) [25]. The Cvi partial resistance to PPV was also mapped to the *rwm1* locus and linked to the Cvi *cPGK2* allele carrying the S78G mutation. However, it was intriguing that Columbia (Col-0) emerged from the GWAS as being partially resistant to PPV, although the S78G substitution is absent from the Col-0 *cPGK2* allele. Poque et al. [25] suggested that a lower expression of *cPGK2* in Col-0 compared to the susceptible *Ler* could explain its partial resistance to PPV. Collectively, these studies present strong evidence that viruses can recruit the glycolytic enzyme PGK in order to establish their infection cycle.

## 7. PGKs Emerging Role for Viruses: An Energy-Producing Enzyme during Virus Cycle

Baker’s yeast (*Saccharomyces cerevisiae)* has long been used as a model to study the functions of host proteins in the viral infection cycle. The discovery that it can support infection by a plant tombusvirus, *Tomato bushy stunt virus* (TBSV), opened new avenues for the characterization of host susceptibility factors [44]. A yeast 2 hybrid screening identified the yeast cytosolic PGK1 as a direct interactor with TBSV replication proteins [45]. The co-option of PGK for the TBSV replication cycle was also confirmed in planta in *N. benthamiana*: plants knocked down for a cytosolic PGK showed low virus accumulation in inoculated leaves, but in an interesting development, the authors used a FRET-based method that locally estimates ATP levels in the cell. By doing so, they revealed a decrease in ATP in the viral replication complex (VRC) of *N. benthamiana* TBSV-infected knockdown plants [22,44]. The authors concluded that the virus recruited the PGK enzyme in order to produce energy in the form of ATP required for Hsp70 chaperone activity during virus multiplication. The co-opted Hsp70 proteins aid the viral replicase complex assembly and the activation of the viral polymerase during TBSV replication [22]. It would be interesting to see whether a similar role could be proved for potyviruses, by which a potyviral partner could recruit a PGK for a similar purpose, keeping in mind that, as a notable difference from the *Arabidopsis* cPGK2 acting in the virus cycle, the yeast PGK1 is cytosolic.

Therefore, assuming that the role of PGK would be similar to feed the virus with ATP, an exciting prospect is to understand how this happens in the plant cell. The easiest way would be for the virus to recruit a cytosolic PGK; hence, PGK3 would be the ideal target, although one can imagine that a small subset of cPGK2 could remain cytosolic through alternative translation initiation or alternative splicing (Figure 5, pathway II). The roles of the chloroplast have been widely evoked in the potyvirus infection cycle [46,47,48,49]. Chloroplast involvement in the *Turnip mosaic virus* (TuMV) cycle has been demonstrated in model plants and crops. The TuMV 6K2 viral proteins, loaded in the endoplasmic reticulum (ER) membrane-formed vesicles, targets and associates to the chloroplast membrane to realize a successful infection cycle [50]. PPV-infected susceptible pea (*Pisum sativum*) and apricot (*Prunus armeniaca*) plants showed a burst of oxidative stress, leading to an increased level of reactive oxygen species (ROS), which have a deleterious effect for plants [46,48]. Moreover, a strong reduction of antioxidant enzymes, as well as a damaged chloroplast ultrastructure, are observed in infected plants. This viral-induced chloroplast phenotype is strikingly similar to the chloroplasts of the *cpgk1* × *cpgk2* mutant [38]. Further investigations should clarify why potyviruses preferentially target chloroplasts and why their located factors are confined by viruses during the infection cycle. Collectively, these data suggest that the recruitment of host enzymatic factors is a viral strategy to shut off plant defences by attacking organelles like chloroplasts, leading to plant death [51]. In light of these events, it would be tempting to speculate that following initial infection, chloroplast-imported cPGK2 could leak out from the chloroplast, where they could be recruited in the VRCs (Figure 5, pathway III).

In conclusion, the plant PGK family exhibits very exciting susceptibility factors to a range of RNA viruses. However, it can be noted that 10 years after their discovery, they have not emerged as a major class of resistance genes characterized in crops, which suggest that they may not have been selected during crop domestication. Nevertheless, we propose that they could be very interesting targets to develop resistances through biotechnology approaches. What has been learned about susceptibility factors eIF4E in recent years could help direct future studies to better understand how viruses use PGKs, as well as how to turn them into successful genetic resistances. We addressed **5 outstanding questions** based on the current status of PGKs knowledge to potentially develop, or engineer through genome editing, PGKs resistant alleles aimed to contain potyviruses.

1-Do PGKs perform redundant functions in crops, as has been shown in *Arabidopsis thaliana*?2-Do viruses use either plant cytosolic and/or chloroplastic PGKs to realize their infection cycle?3-How do viruses proceed to recruit plant PGKs?4-Are the Cvi ecotype’s resistance to WMV and PPV associated to the cPGK2 gene polymorphic? What could be the mechanism of cPGK2-based resistance in the Cvi ecotype?5-Can we design, using CRISPR strategy, PGKs-resistant alleles based on PGKs natural resistant allele polymorphism?

## 8. Conclusions: PGKs as Potential New Genetic Resistances to Cope with Viruses

Converging data from yeast, *Arabidopsis*, and tobacco strongly suggest that PGKs could be an additional target for developing loss of susceptibility resistance to potyviruses and other ssRNA+ viruses, with a strong potential for translational biology. Interestingly, this prospect is likely to meet the same challenges faced in the development of eIF4E-based resistance, such as the role of gene redundancy towards the resistance spectrum and resistance durability, and the necessity to balance resistance with normal plant development. An additional challenge will be to determine the role of the PGKs subcellular localization for the enzyme to be recruited by the virus, as in the yeast and *Arabidopsis* models, plant viruses can either use cytosolic or chloroplastic PGKs, respectively, to perform their infectious cycle [22,23]. Based on the exhaustive analyses of *Arabidopsis* PGKs redundancy towards metabolism and development [27], a similar study assessing how viruses can distinguish and hijack PGKs proteins would be the next step to perform. Therefore, it is likely that before being able to embark on developing genetic resistance, a few important points must be sorted out (see **Outstanding Questions, above**).

Based on these studies and a close analysis of the PGKs genes redundancy for a given crop, resistance strategies can be elaborated. The general knowledge acquired on *Arabidopsis* could be extended in order to decide which gene should be targeted, and how this should be accomplished (inactivation versus modification). Allele mining can be a powerful strategy to find the allele of choice, with specific mutations. However, genome editing strategies have emerged in recent years as a tool of choice that seems to be particularly fitted for resistance to pathogens [52]. Due to the high versatility of CRISPR-based technologies, it is possible to target a specific region of the genome through the expression of a guide RNA and a nuclease Cas9, generating insertion or deletion (indels) that could result in gene loss-of-function. As for *eIF4E*-based resistance to potyviruses, it can be expected that gene inactivation could give rise to at least partial resistance [21]. As seen above for the functional analysis of PGKs genes in *Arabidopsis*, it can be expected that inactivating single genes will not significantly affect the plant development, while targeting several genes at the same time could lead to severe developmental defects. This completely mirrors the situation for the *eIF4E* gene family in tomato and in *Arabidopsis* [53]. If this is the case, the gene edition by targeting candidate mutations, such as S78G or residues involved in substrate binding (Figure 1), could be performed in order to create a functional allele conferring resistance: a similar rationale drove the development of an *eIF4E1* resistance allele through base editing, causing a single amino acid change in the eIF4E1 protein [54]. Finally, recent development of the CRISPR technology could be very relevant for modulating the expression of PGKs: promoters could be randomly edited to modify the gene expression, as well as lower its expression, as shown in tomato plants through random targeting of the cis-element [55]. This could be a way of generating partial resistance, as the low expression of the *cPGK2* Col-0 allele is associated with partial resistance to PPV [25], without impacting the plant development. Finally, experiments performed using *Arabidopsis* have shown that deletion at the C- and N-terminal of TP of ferredoxin greatly impacted the protein targeting inside the chloroplast, which could be applied to relocate PGKs in the cell [56]. Establishing a combination of mutations simultaneously in different PGKs could be achieved through multiplexing, in order to avoid viruses that might hijack several PGKs at the same time. This should help enlarge both the resistance spectrum and its durability. It can be expected that PGKs are ubiquitous as susceptibility factors to plants viruses. If so, their high level of conservation among crops could be a prospect to generate resistances in a large number of plants in which PGKs could promote virus multiplication. To achieve this goal, we need to know which PGKs alleles to combine, or which metabolic enzyme-encoded genes to edit together in plants in order to produce resistance against viruses.

Further investigations identified other glycolytic enzymes as pro-viral factors in yeast and in plants: the induction of pyruvate decarboxylase 1 (PDC1) and alcohol dehydrogenase 1 (ADH1) expression, lowly expressed in tobacco leaves, are significantly increased following TBSV infection. Moreover, both PDC1 and ADH1 proteins interact directly with p33 and p92 viral replication proteins [57]. Similar to PGK1, PDC1 and ADH1 are therefore also co-opted by TBSV to fill the energy requirement of virus replication: indeed, a higher level of ATP is measured inside the VRC [22,44]. These data collectively illustrate the dependence of the single-stranded positive RNA viruses replication cycle on glycolytic enzymes, indicating further ways to foster resistance development for crops.

## Figures and Tables

**Figure 1 viruses-14-01245-f001:**
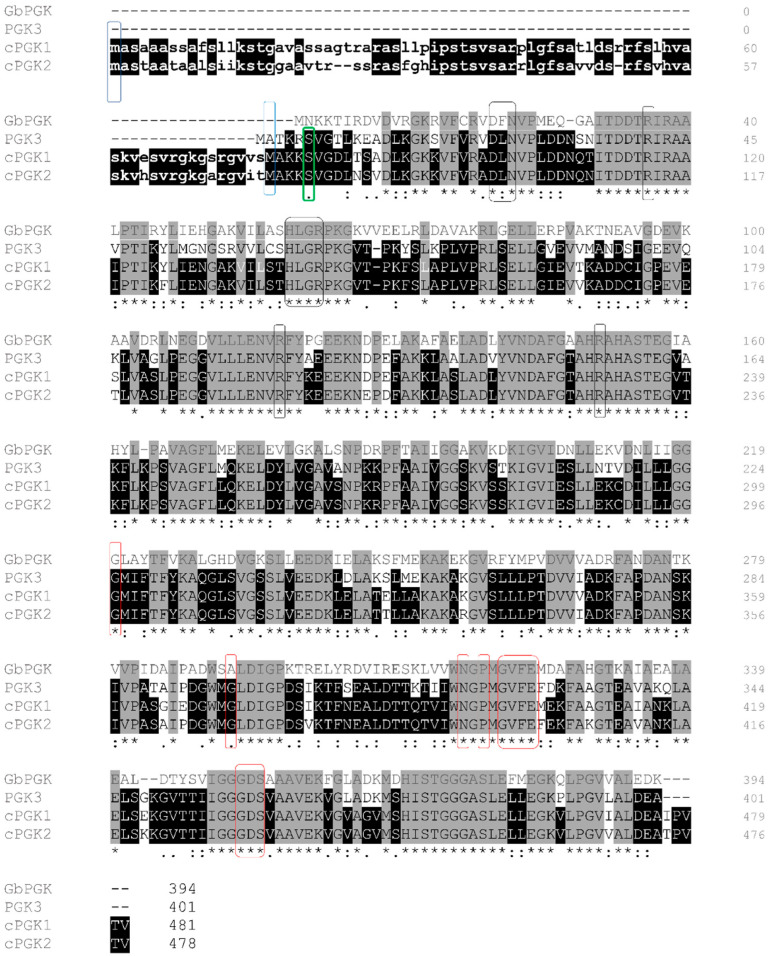
Sequence alignment of *Arabidopsis thaliana* and *Geobacillus stearothermophilus* PGKs protein sequences show a high sequence identity between plants and bacterial PGKs. *Arabidopsis* cPGK1 (At3g12780), cPGK2 (At1g56190), and PGK3 (At1g79550) are aligned with the bacillus PGK (P18912) using MUSCLE (https://www.ebi.ac.uk/Tools/msa/muscle/, accessed on 18 June 2021). Amino acid sequences of the chloroplastic isoforms begin with a transit peptide (TP, in lowercase) that starts with methionine 1. Methionine 2 (blue boxed methionine) is present in cPGK1 (Met77) and cPGK2 (Met74) sequences in the first exon. Conserved residues between the 4 sequences are shown in the grey background, and conserved residues between the *Arabidopsis* PGKs are shown in the black background. Amino acid residues involved in 3-PGA binding are black boxed and those involved in ADP binding are red boxed. Conserved S78 residue substituted by G78 in the Cvi ecotype displaying resistances to potyviruses [24] is green boxed.

**Figure 2 viruses-14-01245-f002:**
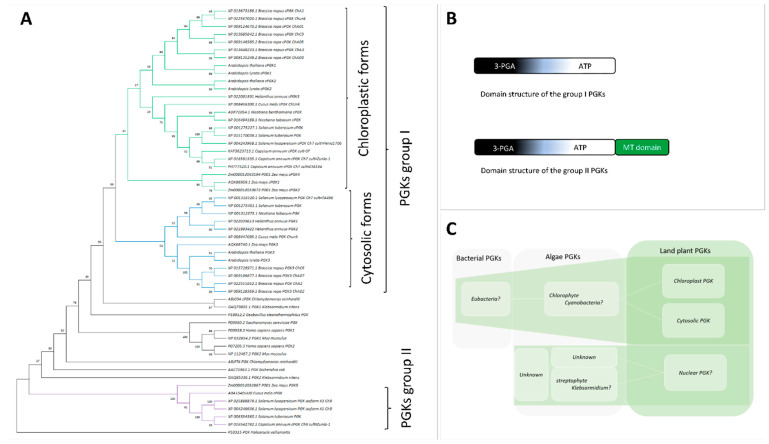
**Maximum likelihood tree of 55 PGKs protein sequences reveals 2 distinct groups of PGKs in the plant**. (**A**) The phylogenetic tree obtained with MEGA version 11 using the PGK of *Haloarcula vallimortis*, an archaea, as the root. The protein sequences are retrieved from GenBank (https://www.ncbi.nlm.nih.gov/genbank/, accessed on 24 October 2021), maize (https://www.maizegdb.org/, accessed on 24 October 2021), and cucurbits database (http://cucurbitgenomics.org/organism/3, accessed on 24 October 2021) for a conservation analysis. The retrieved sequences are aligned with MUSCLE (https://www.ebi.ac.uk/Tools/msa/muscle/, accessed on 26 October 2021). Subsequently, using protTest software V3.4.2 (https://github.com/ddarriba/prottest3/releases, accessed on 25 October 2021), the WAG matrix is predicted as the best model of amino acids substitution. The obtained tree shows 2 groups (group I and group II) of PGKs that likely have a distinct origin. Group I contains plant cytosolic and chloroplastic PGKs coloured in blue and green, respectively. This group is closely related to *Chlamydomonas reinhardtii* chloroplastic PGK, which might be its ancestor [29]. A second class of PGK, in purple colour, forms group II, showing a striking divergence from group I. The members of this group all have a predicted methyltransferase (MT) domain (IPR029063) [30]. *Klebsormidium nitens* has a chloroplastic PGK (GAQ85336.1) displaying a homologous structure domain, as opposed to the plant group II PGKs. PGKs from others organisms are black-coloured. Organism names are preceded by the protein accession numbers for each protein sequence. (**B**) Illustration diagrams of domains found in group I and group II plant PGKs. 3-PGA (in black background) and ATP (in white background) represent sites found in the PGKs group I and group II. An additive methyltransferase domain (MT domain) is also observed in group II PGKs. (**C**) Proposed evolution model of the described plant PGKs based on our phylogenetic analysis.

**Figure 3 viruses-14-01245-f003:**
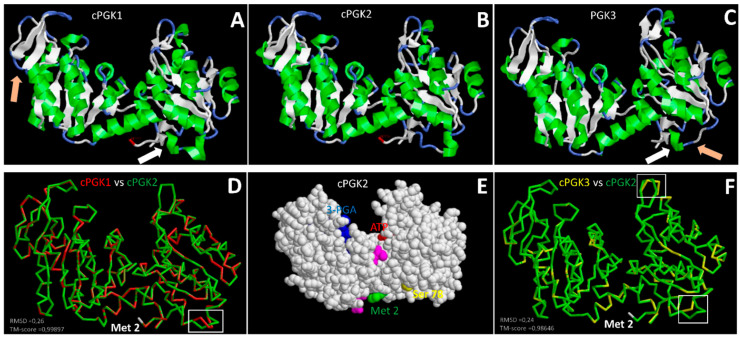
Predicted 3D structures and comparison of *Arabidopsis thaliana* phosphoglycerate kinases (PGK) reveal a high homology between chloroplastic and cytosolic PGKs isoforms. (**A**–**C**) Ribbon structures of the *Arabidopsis* cPGK1, cPGK2, and PGK3 enzymes obtained on Swiss model (https://swissmodel.expasy.org/interactive accessed on 28 July 2021) using the bacillus PGK having a crystallized structure as template. In (**A**–**C**), green and white colours, respectively, show helices and sheets in cPGK1, cPGK2, and PGK3 ribbon diagrams. Taking cPGK2 as a reference for structure model comparison, orange arrows in cPGK1 and PGK3 point to absent helixes that are present in cPGK2, while white arrows points to helixes absent in cPGK2, but present in cPGK1 and PGK3. Superimposed backbones of cPGK2 (green) with cPGK1 (red) and PGK3 (yellow) in (**D**,**F**), respectively, using TM-align (https://zhanggroup.org/TM-align/ accessed on 29 July 2021): RMSD and TM score values that are parameters evaluating the structural homology of proteins are indicated in each pairwise comparison. The few depicted structural dissimilarities between the isoforms corresponding to non-superimposed parts of the compared proteins are white-boxed in (**D**,**F**). (**E**) Molecular surface of cPGK2 enzyme using Raswin software [37] Version 2721 where structural information showing, in blue and red colours, residues involved in 3-PGA and ADP binding site; green, pink, and yellow colours point, respectively, to methionine 2 (position 74 from methionine 1), the hinge region that links the 2 domains, and Serine 78, a candidate mutation involved in the Cvi ecotype resistance against *Watermelon mosaic virus* (WMV).

**Figure 4 viruses-14-01245-f004:**
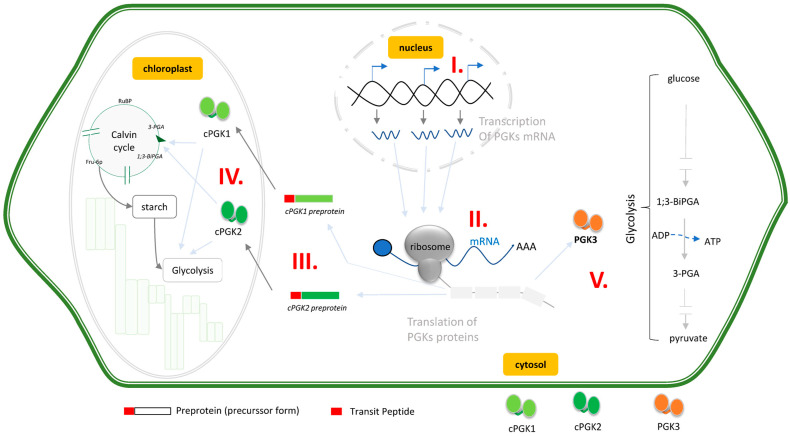
*Arabidopsis* phosphoglycerate kinases (PGK) localizations and functions in plant cell. *Arabidopsis* have 3 PGKs genes located in the nucleus (I) and encode two chloroplastic forms (IV, cPGK1 and cPGK2) and a cytosolic form (V, PGK3) [27]. Chloroplastic forms are translated in precursor proteins (III), which is subsequently targeted in the chloroplast, thanks to its N-terminal transit peptide (TP, red rectangle). These forms will be processed after chloroplast translocation to yield mature proteins (IV) without the transit peptide [39]. cPGK1 (light green) and cPGK2 (dark green) are assumed to be involved in the Calvin cycle and in chloroplast localized glycolysis. PGK3 (V), which is cytosolic, is assumed to be the glycolytic isoform [27] involved in cytosolic glycolysis, a sequential breakdown of glycose by numerous enzymes, yielding pyruvate. The compounds produced in glycolysis and the Calvin cycle are: RuBP: ribulose biphosphate; 3-PGA: phosphoglycerate kinase; 1,3-BiPGA: 1,3-biphospoglycerate; Fru-6p: fructose 6-phosphate.

**Figure 5 viruses-14-01245-f005:**
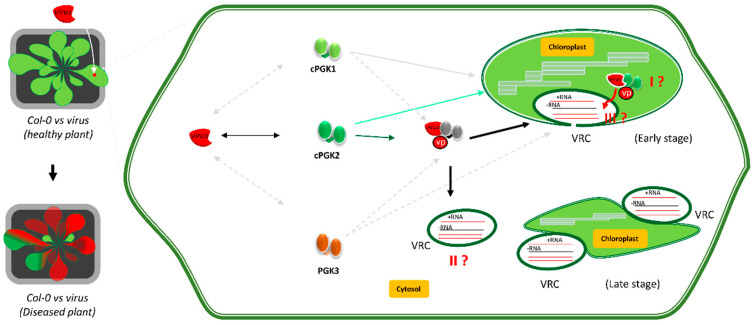
Illustration model of a potyvirus replicating in susceptible *Arabidopsis thaliana* plants, using the phosphoglycerate kinases (PGK) as host factors and the chloroplast membrane as a replication site in the early (normal chloroplast) and late (damaged chloroplast) infection stages. The three *Arabidopsis* PGKs are presented: cPGK1, cPGK2, and PGK3, with the *Watermelon mosaic virus* (WMV), which is known to infect *Arabidopsis* through cPGK2. Three hypotheses are shown: I—WMV is transported in the chloroplast, where it recruits the targeted cPGK2. II—cPGK2 is hijacked in the cytosol, before being targeted to the chloroplast, or because some of it is targeted to the cytosol through alternative splicing or translation through Met2. III—at the later stage of the infection, and following damages in the chloroplast, the chloroplastic cPGK2 leak out from the chloroplast to the viral replication complexes (VRC).

**Table 1 viruses-14-01245-t001:** Summary of phenotypic analyses of the current *Arabidopsis* PGKs mutants. The expression analysis of PGKs genes in each of these lines reveals that the *cpgk1* (*GK_172A12*; *GK_908E11*) lines are knock down, while *cpgk2* and *pgk3* (*SALK062377*; *SALK066422*) lines are complete knockout [27].

Genotypes/Agi/Type of Mutant	Phenotypes	References
cpgk1 (GK_172A12; GK_908E11)	Mild phenotype, lower photosynthetic capacity, lower starch content, metabolite content variation.	[27]
cpgk1 CRISPR-Cas9 line	Normal, similar to the wild-type plants.	[36]
cpgk2 (SALK016097) ^1^	^1^ The albino phenotype is not caused by cPGK2 loss-of-function.	
cpgk2 CRISPR-Cas9 line	Normal, plants slightly smaller than wild-type plants.	[23,26,36]
pgk3 (SALK062377; SALK066422)	Mild phenotype, reduction in growth parameters, higher starch level, metabolite content variation.	
cpgk1 × cpgk2 (CRISPR-Cas9 line)	Albino plants that die, defective chloroplast development, phenotype can be partially restored by exogenous sugar.	[36]
cpgk1 × pgk3 (T-DNA line)	Reversion of the smaller phenotype caused by pgk3.	[26] Not available.
cpgk2 × pgk3	Not available.	

^1^ The lethal phenotype of the cPGK2 (SALK016097) T-DNA mutant is not associated with specific cPGK2 insertion.

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
