# Peer review of "Exploring New Routes for Genetic Resistances to Potyviruses: The Case of the Arabidopsis thaliana Phosphoglycerates Kinases (PGK) Metabolic Enzymes"

_viruses, 2022, doi:10.3390/v14061245_

Round 1
Reviewer 1 Report
The article of Diop and Gallois reviews information available on phosphoglycerate kinases (PGK), which, in addition to their important role in cell metabolism, act as susceptibility factors for several plant viral infections. Although only two of its seven sections deal directly with the proviral function of these proteins, I believe the review is of general interest for virologists. Moreover, the article is well-organized, clear and provides up-to-date information.
My main criticism relates to the fact that while a very detailed information is provided about PGKs of Arabidopsis, almost nothing is explained about the PGKs of Nicotiana benthamiana, the other plant in which these proteins have shown to play a role in viral infections. How many PGKs does N. benthamiana have? Why only one of them is included in Figure 3? How specific is the contribution of a PGK for BaMV and TBSV infection in N. bentamiana? This is especially important because targeting susceptibility factors to engineer antiviral resistance is easier when there is redundancy in the plant function but specificity in the proviral one (as appears to be, at least partially, the case for the arabidopsis PGKs). On the other hand, the article suggests that PGKs may be helping viral infection with an activity parallel to that they play in plant metabolism (contributing to ATP supply). In this regard, it would be interesting to explain what it is known about the requirement of the enzymatic activity of PGKs for BaMV, TBSV and potyvirus infections.
Other minor comments:
- line 15: crops are plants
- line 38: What is the reason of selecting these particular genes?
- lines 44 and 45: I would suggest to include other well-studied susceptibility factors, as TOM proteins (Yamanaka, T., et al. 2000.PNAS 97: 10107; Tsujimoto, Y., et al. 2003. EMBO Journal 22: 335).
- line 79: There are three homologous PGKs (line 87). I would suggest to replace “two homologous PGKs” by something like “two forms/types of homologous PGKs”
- lines 93, 109, 114, 186, and many other places: Homology is a non-quantifiable evolutive trait (there is homology or there is not). This term should be replaced by “identity”, “similarity” or something like that.
- line 122: Genericity, or generality?
- line 131 (Figure S1): I would suggest pointing out in Figure S1 residues involved in 3-PGA binding and ADP binding (as in Fig. 1), as well as the mT domain of atypical PGKs.
- line 180-1: Although the function of atypical PGKs is intriguing, I suppose that it has been demonstrated that they have PGK enzymatic activity, am I right? Is it known if they have methyl transferase activity?
- line 223: White with lowercase first letter
- Table 1, line 7: Mild reduction or Mild phenotype, reduction in growth …?
- line 334: Which is the BaMV protein that interacts with the chloroplastic PGK? Is this interaction with the full-length PGK or with the protein that results from the signal peptide removal?
- lines 336 and 384: Are these knocking-downs expected to be specific for single PGKs?
- line 339: Figure 5 specifically refers to potyviruses.
- line 367: Either “potyviruses replicating” or “a potyvirus replicating”.
-lines 386-7: Has any PGK been found colocalizing with replication complexes in virus-infected plants?
- line 392: Is it known if the PGK involved in TBSV infection in N. benthamiana is also cytosolic?
- line 434: Why only potyviruses? Given that PGKs are involved in the replication of viruses of three different families, is it not expected that they could be a general cofactor of plant virus infections? And, if this is true, are not PGKs potential new sources of resistance against plant viruses beyond potyviruses?
- line 437: The word “viruses” is redundant; “ssRNA+” is an abbreviation of “single-stranded positive RNA virus” (line 58).
Reviewer 2 Report
Line 18, ‘Arabidopsis’ should be italicized in the manuscript.
Line 42, ‘translating’ is changed to ‘translate’
Line 65, ‘extend’ is changed to ‘extends’
Line 94, ‘and PGK3’ are changed to ‘, and PGK3’
Line 95, ‘Amino Acids’ are changed to ‘Amino acids’
Line 113, ‘closest’ is changed to ‘closer’
Line 115, ‘PGK, a PGK whose’ are changed to ‘PGK whose’
Line 129, ‘twenty-four chloroplastic forms and’ are changed to ‘twenty-four chloroplastic forms, and’
Line 144, ‘comforts’ is changed to ‘conforms’
Line 157, ‘Chlamydomonas reinhardtii’ should be italicized.
Line 160, ‘klebsormidium nitens’ are changed to ‘Klebsormidium nitens’
Line 178, ‘Cucurbitaceae and’ are changed to ‘Cucurbitaceae, and’
Line 181, ‘the function of the atypical’ are changed to ‘, the function of the atypical’
Line 192-193, ‘bind respectively 3-PGA 192 and ATP’ are changed to ‘bind 3-PGA 192 and ATP, respectively’
Line 216, ‘Arabidopsis thaliana’ should be italicized.
Line 249, ‘RNAi’ is changed to ‘RNAi, ’
Line 280, ‘TDNA’ is changed to ‘T-DNA’
Line 298, ‘cPGK2 cannot be’ are changed to ‘cPGK2, can’t be’
Line 312-313, The sentence ‘While cPGK2 on its own can support the plant 312 development, PGK3 cannot.’ should be rewritten.
Line 367, ‘Arabidopsis thaliana’ should be italicized.
Line 379, ‘infectious cycle’ are changed to ‘infection cycle’
Line 380, remove the word ‘the’
Line 393, ‘similarly’ is changed to ‘similar’
Line 430-431, ‘polymorphism’ is changed to ‘polymorphic’
Line 443, ‘used’ is changed to ‘use’
Line 460, ‘Amino Acid’ are changed to ‘amino acid’
Line 478, ‘enlarging’ is changed to ‘enlarge’
Line 483, ‘resistant’ is changed to ‘resistance’
Line 488, ‘similarly’ is changed to ‘similar’
Lines 496-498, ‘animals, algae and bacteria’ are changed to ‘animals, algae, and bacteria’
Lines 571, 602, 627, and 642, please provide the name of these journals.
Round 2
Reviewer 1 Report
The new version of the manuscript approaches satisfactorily the main concerns raised in the first review